# DIVER 🤿: LARGE LANGUAGE MODEL DECODING WITH SPAN-LEVEL MUTUAL INFORMATION VERIFICATION

## ABSTRACT

Large language models (LLMs) have shown impressive capabilities in adapting to various tasks when provided with task-specific instructions. However, LLMs using standard decoding strategies often struggle with deviations from the inputs. Intuitively, compliant LLM outputs should reflect the information present in the input, which can be measured by point-wise mutual information (PMI) scores. Therefore, we propose DIVER, a novel approach that enhances LLM **D**ecoding through span-level PM**I** **VER**ification. During inference, DIVER first identifies divergence steps that may lead to multiple candidate spans. Subsequently, it calculates the PMI scores by assessing the log-likelihood gains of the input if the candidate spans are generated. Finally, the optimal span is selected based on the PMI re-ranked output distributions. We evaluate our method across various downstream tasks, and empirical results demonstrate that DIVER significantly outperforms existing decoding methods in both performance and versatility.

## 1 INTRODUCTION

The emergence of large language models (LLMs) has significantly reformed the paradigms in natural language processing (NLP) (Brown et al., 2020; Anil et al., 2023; Touvron et al., 2023). With instruction-tuning (Ouyang et al., 2022; Zhang et al., 2023b) or in-context learning (ICL) (Brown et al., 2020; Dong et al., 2022), LLMs yield impressive performance on various downstream tasks. Despite the strong versatility, LLMs pre-trained with unsupervised corpora using language modeling as the training objective frequently generate content unfaithful to inputs in particular downstream tasks (Bang et al., 2023; Rawte et al., 2023; Guerreiro et al., 2023). For example, in machine translation (MT), LLMs may generate irrelevant additional content or overlook important parts of the original inputs (Zhang et al., 2023a). Such issues would affect the outputs of LLMs, decreasing the reliability of deployment in practical scenarios.

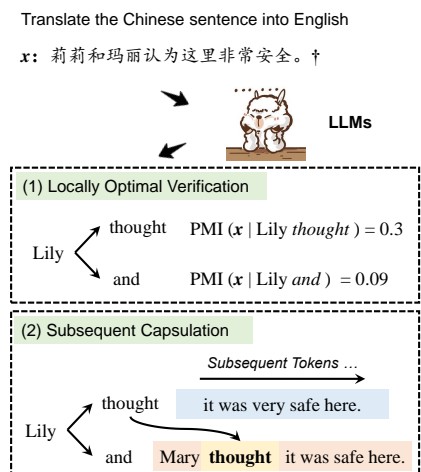

Figure 1: The illustration about verification based on the disparity of a single token may lead to a locally optimal outcome.

Intuitively, compliant LLM outputs should follow instructions and accurately reflect the information present in the source inputs. Therefore, a direct solution is to verify whether the candidate tokens at each decoding step have a strong correlation with the input, which can be measured by point-wise mutual information (PMI) (Church & Hanks, 1990) between the candidate token $y_i$ and the input $x$. However, when the input sequence $x$ contains abundant information, the disparity in the amount of information between $y_i$ and $x$ is significant, making such

a verification less effective. As illustrated in Figure 1[1], verification with inadequate information may bring a local optimum at the current decoding step, diverting from achieving globally optimal results like (1). However, if the LLM generates *and*, *thought* can also appear in subsequent tokens (subsequent encapsulation (2)), potentially leading to a better translation. We believe that effectively addressing this concern entails harnessing sufficient information for PMI calculation, thus enhancing the probability of obtaining a better output.

Based on the above consideration, we propose DIVER, enhancing LLMs **D**ecoding via span-level PM**I** **VER**ification. Specifically, at the decoding step with multiple candidate tokens (divergence point), LLMs generate several continuous spans started by these candidate tokens. Subsequently, DIVER selects the continuous token span by concurrently assessing the probability at the divergence point along with PMI scores between continuous spans and the input text. Specifically, through equivalent transformation, PMI scores can be converted into the calculation of log-likelihood gains of the input if the spans are generated. With the help of span-level PMI verification, DIVER can encourage LLMs to generate accurate and coherent outputs.

We evaluate DIVER on various downstream tasks, including code generation, dialogue response generation, element-constrained generation, knowledge question answering, machine translation, text summarization as well as story generation. Compared to vanilla decoding methods such as greedy decoding or nucleus sampling (Holtzman et al., 2020), and advanced contrastive decoding strategies (Li et al., 2023; Shi et al., 2023), DIVER consistently achieves substantial performance enhancements across multiple tasks, demonstrating its effectiveness and versatility.

## 2    BACKGROUND - LLM DECODING

In the era of LLMs, natural language tasks transition into open-ended language generation scenarios, where inputs serve as part of prompts, driving LLMs to generate continuations in an auto-regressive manner. Given the input $x = \{x_1, x_2, \cdots, x_n\}$, the output token $y_i$ is selected based on the probability conditioning on the preceding tokens.

$$y_i \sim \log p(y_i|y_{<i}, x) \tag{1}$$

The commonly used decoding method is greedy search or nucleus sampling. Specifically, greedy search chooses the token with the largest probability according to the distribution at each decoding step. Nucleus sampling, on the other hand, samples from the top-$p$ percentile of the distribution, thereby enhancing the diversity of the generated context. However, using either greedy search or nucleus sampling may cause LLMs to generate outputs that are unfaithful to the inputs, resulting in hallucination problems (Rawte et al., 2023; Ji et al., 2023; Huang et al., 2023b).

## 3    OUR METHOD

### 3.1    DIVER 🤿 - DECODING WITH POINT-WISE MUTUAL INFORMATION VERIFICATION

To alleviate the unfaithful issue, we strengthen the correlation between the input $x$ and the ongoing generated token $y_i$ via point-wise mutual information (PMI). At decoding step $i$, $y_i$ is controlled by the generated tokens $y_{<i}$ and influences the succeeding tokens $y_{>i}$. Therefore, we argue that the selection of $y_i$ should consider both the original output distribution and the overall PMI score between $x$ and $y$:

$$y_i \sim \log p(y_i|y_{<i}, x) + \text{PMI}(y, x) \tag{2}$$

Because $y_{<i}$ have already been generated, $\text{PMI}(y, x) \propto \text{PMI}(y_{\geq i}, x|y_{<i})$. $\text{PMI}(y_{\geq i}, x|y_{<i})$ refers to the PMI score between $x$ and $y_{\geq i}$, conditioned on $y_{<i}$. Thus, equation (2) can be rewritten as:

$$y_i \sim \log p(y_i|y_{<i}, x) + \text{PMI}(y_{\geq i}, x|y_{<i}) \tag{3}$$

Regrettably, $\text{PMI}(y_{\geq i}, x|y_{<i})$ can only be computed when the tokens are completely generated. It will significantly increase the computational cost and decrease the inference speed. To avoid this

---

[1]The standard reference for the input $x$ is *Lily and Mary thought it was very safe here.*

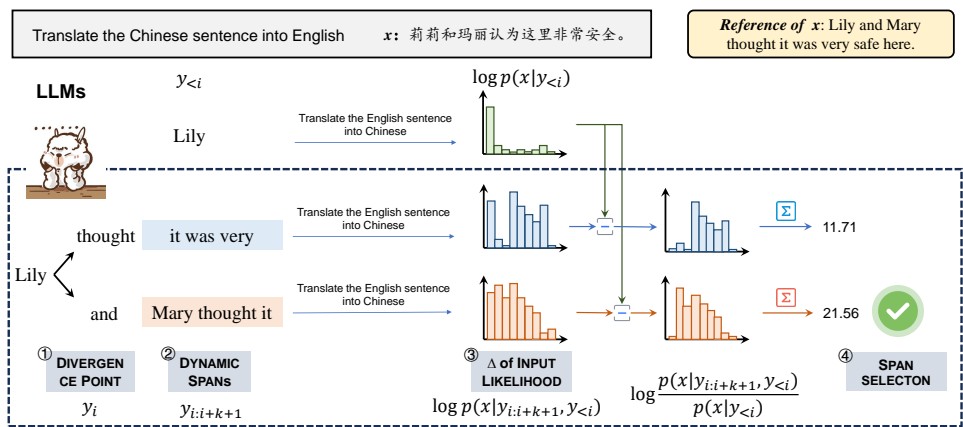

Figure 2: An overview of DIVER. It first identifies the divergence points and generates several candidate spans. Then, it computes the delta $\Delta$ of the log-likelihood of input $x$ (PMI scores) for the distribution re-ranking. Finally, a token span is selected based on the re-ranked distribution.

issue, we request that the model generate the next $k$ tokens, denoted as $y_{i:i+k+1}$, rather than the entire sequence for $y_i$ selection:

$$y_i \sim \log p(y_i|y_{<i}, x) + \text{PMI}(y_{i:i+k+1}, x|y_{<i}) \tag{4}$$

Given that $y_i$ determines subsequent tokens and $y_{i:i+k+1}$ have already been generated for PMI calculation, selecting a candidate span $y_{i:i+k+1}$ instead of a single token $y_i$ can further reduce the computational cost. This operation can achieve a balance between decoding quality and speed:

$$y_{i:i+k+1} \sim \log p(y_i|y_{<i}, x) + \text{PMI}(y_{i:i+k+1}, x|y_{<i}) \tag{5}$$

Based on the definition of PMI, equation (5) can be written as:

$$y_{i:i+k+1} \sim \underbrace{\log p(y_i|y_{<i}, x)}_{\text{vanilla distribution}} + \underbrace{\log \frac{p(x|y_{i:i+k+1}, y_{<i})}{p(x|y_{<i})}}_{\text{PMI verification}} \tag{6}$$

Specifically, the verification part can be viewed as the likelihood gains of the input when $y_{i:i+k+1}$ is decoded, which can be computed via backward teacher-forcing decoding[2]:

$$\log \frac{p(x|y_{i:i+k+1}, y_{<i})}{p(x|y_{<i})} = \log \frac{\prod_t p(x_t|y_{<i+k+1}, x_{<t})}{\prod_t p(x_t|y_{<i}, x_{<t})} = \sum_t \log \frac{p(x_t|y_{<i+k+1}, x_{<t})}{p(x_t|y_{<i}, x_{<t})} \tag{7}$$

Therefore, the PMI enhanced span selection distribution $q(y_{i:i+k+1}|x, y_{<i})$ can be written as:

$$q(y_{i:i+k+1}|x, y_{<i}) = \log p(y_i|y_{<i}, x) + \sum_t \log \frac{p(x_t|y_{<i+k+1}, x_{<t})}{p(x_t|y_{<i}, x_{<t})} \tag{8}$$

## 3.2 DIVER FOR LLMS

Figure 2 illustrates the basic process of DIVER adapted for LLMs. Initially, DIVER identifies the **DIVERGENCE POINT**, where several potential candidate tokens may emerge at decoding steps. Once identified, DIVER requests LLMs to generate **DYNAMIC SPAN**s as candidates and calculates the PMI scores. These scores are then used to re-rank the vanilla distributions for **SPAN SELECTION**.

---

[2]Several methods can be adopted for computing the backward log-likelihoods, such as using models fine-tuned on data from $y \rightarrow x$. However, for the sake of simplicity, we use the same LLM throughout this work unless otherwise specified.

**DIVERGENCE POINT** Considering that the tokens predicted with high confidence are typically less prone to error (Guo et al., 2017; Zhu et al., 2023), we borrow the approach proposed in (Li et al., 2023) to detect the positions that might lead to inaccurate decoding. Meanwhile, we truncate the candidate set $\mathcal{C}(i)$ accordingly:

$$\mathcal{C}(i) = \{y_i \in \mathcal{V} | p(y_i|y_{<i}) \geq \gamma \max_{w \in \mathcal{V}} p(w|y_{<i})\} \tag{9}$$

where $\mathcal{V}$ is the vocabulary and $\gamma$ is the hyper-parameter to control the truncating range.

For the decoding steps with multiple candidate tokens ($|\mathcal{C}(i)| > 1$), LLMs are typically not confident in the output distribution. All the top tokens can be suitable for the current step, and each token may lead to a diverse sequence. Therefore, we request LLMs to continue generating $k$ tokens, forming several candidate spans.

**DYNAMIC SPAN** In practical experiments, we observe that various tasks exhibit sensitivity to the span length $k$. To address this issue, we introduce an adaptive method for obtaining token spans with dynamic lengths, tailored to specific examples.

For current divergence point $i$ with $\mathcal{C}(i)$ as the candidate token set, LLMs generate succeeding tokens after these candidates and obtain several spans $\{y_{\geq i}^m | 0 < m \leq |\mathcal{C}(i)|\}$. During generation, DIVER records the risk step $r$, which could potentially be the divergence point (as defined in equation (9)) that first emerges within each candidate span. The risk set $\mathcal{R}$ is composed of the first-emerged risk steps $r_m$ in different spans:

$$\mathcal{R} = \{r_m | r_m \leftarrow \min\{j | |\mathcal{C}^m(j)| > 1, j > i\}, 0 < m \leq |\mathcal{C}(i)|\}$$

where $\mathcal{C}^m(j)$ refers to the candidate token set at position $j$ in $m$-th span.

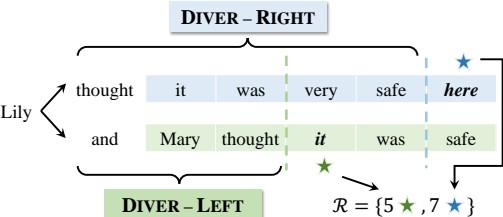

Figure 3: An example illustrates DYNAMIC SPAN acquirement. Bleu and green stars refers to the first-emerged risk points in the two sequences.

Once all first-emerged risk steps in the candidate spans are recorded in $\mathcal{R}$, DIVER pauses generation and utilizes both the LEFT and RIGHT boundaries to calculate the dynamic span length $k$. Figure 3 shows a specific example of DYNAMIC SPAN acquirement. It should be noted that both the LEFT and RIGHT boundaries can form dynamic spans for different examples. Specifically, DIVER-LEFT ensures no omission of any risk point that could lead to divergence but may yield less informative spans, while DIVER-RIGHT ensures sufficient information provision but may select spans containing potential divergence points.

$$\text{LEFT} : k \leftarrow r - i - 1, r = \min \mathcal{R}$$
$$\text{RIGHT} : k \leftarrow r - i - 1, r = \max \mathcal{R}$$

**SPAN SELECTION** After obtaining the DYNAMIC SPANs, DIVER calculates the conditional PMI scores as defined in Equation (7). To achieve this, DIVER first uses a backward instruction, reversing both the output tokens and the input $x$, as illustrated in Figure 2. It then collects and sums the delta of log-likelihood for each token $x_t$ if the candidate token spans are generated, thereby obtaining the PMI scores. Finally, these PMI scores are used to re-balance the distributions according to equation (8). Based on these distributions, DIVER selects candidate spans using either a greedy search or sampling, depending on the task properties.

$$y_{i:i+k+1} \sim \begin{cases} q(y_{i:i+k+1}|x, y_{<i}) & \text{if } y_i \in \mathcal{C}(i), \\ -\infty & \text{otherwise.} \end{cases}$$

After the span selection, DIVER continues decoding from the step $i + k + 1$, repeating the aforementioned steps until it encounters the specified ending tokens.

| Task | Dataset | Evaluation Metrics |
|---|---|---|
| Code Generation | MBPP (Austin et al., 2021) | Pass@1 |
| Machine Translation | Flores-200 (Costa-jussà et al., 2022) | BLEU, 100-TER, BLEURT |
| Text Summarization | CNN/DailyMail (Nallapati et al., 2016) | ROUGE-1/2/L |
| | SAMSum (Gliwa et al., 2019) | ROUGE-1/2/L |
| World-Knowledge QA | Natural Qeustions (Kwiatkowski et al., 2019) | EM, F1 |
| | Web Questions (Berant et al., 2013) | EM, F1 |
| EC Generation | E2E (Novikova et al., 2017) | BLEU, ROUGE-L, NIST, CIDEr |
| | CommonGen (Lin et al., 2020) | BLEU, ROUGE-L, METEOR |
| Dialogue Response | DailyDialogue (Li et al., 2017) | BLEU-1, Distinct-1/2 |
| Story Generation | ROCStory (Mostafazadeh et al., 2016) | BLEU-1, Distinct-1/4 |

Table 1: Datasets and evaluation metrics for various tasks.

| Tasks | Datasets | Basic Decoding | Decoding Methods | | | | |
|---|---|---|---|---|---|---|---|
| | | | Vanilla | CD | CAD | DIVER$_L$ | DIVER$_R$ |
| Dialogue Response | Daily Dialogue | Samping | 16.69 | 16.61 | 17.43 | 17.46 | **18.37** |
| Story Generation | ROCStory | Samping | 37.56 | 37.78 | 38.28 | 37.93 | **38.54** |
| Code Generation$^†$ | MBPP | Greedy | 46.60 | - | 47.73 | 47.93 | **48.67** |
| Translation | Flores-Fr-En | Greedy | 57.86 | 57.29 | 56.18 | **58.69** | 58.60 |
| | Flores-De-En | Greedy | 56.32 | 55.92 | 55.65 | 57.14 | **57.23** |
| | Flores-Bg-En | Greedy | 51.13 | 50.84 | 50.91 | **51.84** | 51.72 |
| | Flores-Zh-En | Greedy | 39.14 | 38.88 | 38.94 | 40.32 | **40.77** |
| | Flores-Ar-En | Greedy | 25.43 | 25.33 | 27.10 | 28.15 | **29.71** |
| Summarization | CNN/DM | Samping | 27.69 | 27.53 | 28.14 | 28.57 | **28.58** |
| | SAMSum | Greedy | 28.87 | 28.32 | 29.49 | 29.78 | **29.82** |
| Knowledge QA | NQ | Greedy | 30.51 | 30.24 | 29.00 | 31.16 | **31.36** |
| | WebQ | Greedy | 34.42 | 34.79 | 34.26 | 35.04 | **35.42** |
| EC Generation | CommonGen | Greedy | 38.22 | 38.44 | 38.21 | **38.61** | 38.13 |
| | E2E | Greedy | 30.75 | 30.29 | 34.60 | 42.34 | **42.52** |

Table 2: Experimental results on various natural language processing tasks with LLaMA-2-7B-Chat. The best scores for each dataset are boldfaced. $^†$ For code generation, we use Code-LLaMA-Instruct-7B for experiments. Because 7B is the smallest model in Code-LLaMA-Family, the CD result is blanked.

## 4 EXPERIMENTS

### 4.1 EXPERIMENTAL SETTINGS

**Task and Datasets**  To demonstrate the versatility of our method, we consider a wide range of language generation tasks. Details are listed in Table 1.

**Models**  We conduct main experiments with LLaMA-2 Family, including LLaMA-2-7B-Chat and LLaMA-2-13B-Chat (Touvron et al., 2023). For specific tasks, like code generation, we respectively use Code-LLaMA-7B-Instruct and Code-LLaMA-13B-Instruct (Roziere et al., 2023) for experiments. To further evaluate the effectiveness of DIVER on other LLMs, we adopt Mistral-7B-Instruct (Jiang et al., 2023), Gemma-7B-Instruct[3], and LLaMA-3-8B-Instruct[4].

**Decoding Methods**  We compare our method with several existing baselines.

\* *Vanilla* refers to using *Greedy Search* or *Nucleus Sampling* with top-$p$=0.90, depending on the task properties.

---

[3] https://ai.google.dev/gemma
[4] https://github.com/meta-llama/llama3

| Tasks | Datasets | Basic Decoding | Decoding Methods | | | | |
|-------|----------|----------------|---------|------|------|--------------------|--------------------|
| | | | Vanilla | CD | CAD | DIVER$_L$ | DIVER$_R$ |
| Dialogue Response | Daily Dialogue | Samping | 16.52 | 17.58 | 17.18 | 17.81 | **18.65** |
| Story Generation | ROCStory | Samping | 37.51 | 37.88 | 38.24 | 38.78 | **38.84** |
| Code Generation[†] | MBPP | Greedy | 54.33 | 51.93 | 53.67 | 55.27 | **55.47** |
| Translation | Flores-Fr-En | Greedy | 59.58 | 59.41 | 59.85 | 59.83 | **60.32** |
| | Flores-De-En | Greedy | 59.07 | 58.40 | 58.92 | 59.04 | **59.16** |
| | Flores-Bg-En | Greedy | 54.24 | 53.69 | 54.56 | 54.43 | **54.82** |
| | Flores-Zh-En | Greedy | 41.75 | 40.91 | 42.04 | 42.44 | **42.69** |
| | Flores-Ar-En | Greedy | 30.27 | 29.37 | 32.68 | 32.69 | **34.15** |
| Summarization | CNN/DM | Samping | 27.89 | 27.69 | 28.06 | 28.20 | **28.27** |
| | SAMSum | Greedy | 30.05 | 29.69 | 30.78 | 30.70 | **30.87** |
| Knowledge QA | NQ | Greedy | 33.43 | 33.76 | 32.83 | 34.52 | **34.72** |
| | WebQ | Greedy | 37.75 | 37.62 | 37.70 | 38.35 | **38.42** |
| EC Generation | CommonGen | Greedy | 40.31 | 40.14 | 40.21 | **41.48** | 41.29 |
| | E2E | Greedy | 34.57 | 35.24 | 39.08 | 42.33 | **48.87** |

Table 3: Experimental results on various natural language processing tasks with LLaMA-2-13B-Chat. The best scores for each dataset are boldfaced. [†] For code generation, we use Code-LLaMA-Instruct-13B for experiments and the CD experiment is performed by using Code-LLaMA-Instruct-7B as the amateur model.

* CD (Li et al., 2023) is contrastive decoding, which selects tokens from the delta distribution between LLMs with the corresponding weaker amateur models[5]. The truncating parameter $\gamma$ for CD is searched from [0.1, 0.3, 0.5, 0.7, 0.9].

$$y_i \sim p(y_i|y_{<i},x) - p_{\text{AMA}}(y_i|y_{<i},x)$$

* CAD (Shi et al., 2023) is context-aware decoding, which makes the contrastive distribution by removing the input $x$. The hyper-parameter $\alpha$ is set as 0.5 as recommended in their paper.

$$y_i \sim (1 + \alpha) \cdot p(y_i|y_{<i},x) - \alpha \cdot p(y_i|y_{<i})$$

* DIVER$_L$ and DIVER$_R$ are our methods, which respectively form the candidate spans by utilizing the LEFT and RIGHT points as boundaries. The hyper-parameter $\gamma$ is set to 0.1 for machine translation and 0.3 for other tasks. Analysis about $\gamma$ is included in section 5.2 [6].

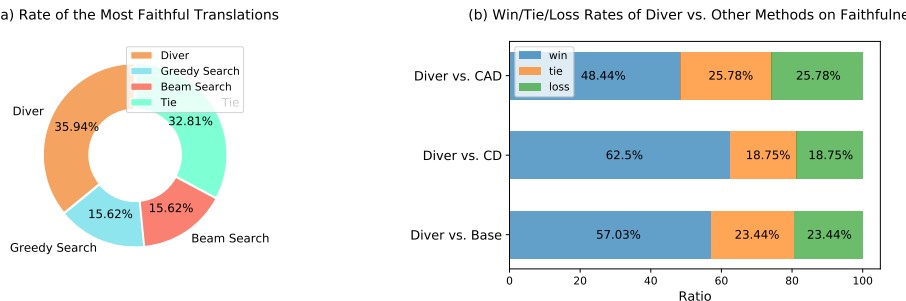

Figure 4: Human judgments on (a) most faithful translation selection among decoding methods in Flores Zh-En and (b) win/tie/loss rates of DIVER compared with other decoding methods in E2E.

---

[5]Unless otherwise specified, we employ Tiny-LLaMA-1.1B-Chat as the amateur model for CD experiments.

[6]It should be noted that CD, CAD, and DIVER are applied on top of basic decoding strategies, either greedy search or nucleus sampling.

## 4.2 EXPERIMENTAL RESULTS

The experimental results are shown in Table 2 and Table 3. Generally, the proposed DIVER achieves the best performance across various downstream tasks. It is worth noting that DIVER$_R$ is slightly better than DIVER$_L$, demonstrating that the amount of information is more essential for verification.

**Machine Translation** For machine translation datasets, the findings reveal that contrastive decoding methods, represented by CD and CAD, fail to yield significant improvements compared to vanilla greedy decoding. Conversely, DIVER consistently surpasses the baseline methods on both 7B or 13B models. Interestingly, the enhancements in performance for similar language pairs are modest, such as Fr-En (+0.83) and De-En (+0.91). However, for distant language pairs like Zh-En and Ar-En, the improvements are substantial, resulting in gains of 1.63 and 4.28 respectively. This underscores the efficacy of the PMI verification strategy for enhancing translations from distant languages to English, particularly those under-represented in LLaMA models.

**Element-Constrained Generation** For this task, DIVER also demonstrates its superiority over other decoding strategies. For E2E, which aims to generate descriptions of restaurants based on given properties, DIVER achieves significant improvements (+11.77 average scores on LLaMA-2-7B-Chat) due to the relatively fixed nature of the references. In contrast, CommonGen requires LLMs to generate logical sentences containing several concepts, with references that are more flexible in expression compared to E2E. Although the improvements are not as significant as in E2E, DIVER still enhances overall performance in CommonGen, achieving a 1.17 average score improvement on LLaMA-2-13B-Chat.

**World-Knowledge QA** For the knowledge QA tasks, we employ in-context-learning (ICL) prompts to constrain the output format, whose demonstration is randomly selected from the validation sets. DIVER further shows its great performance on the QA tasks. We suppose that the reason behind this lies in that the verification boosts the right answer selection by reviewing the relations between entities in questions and candidate answers.

**Summarization, Dialogue Response and Story Generation** These tasks typically allow for significant flexibility in content generation. On one hand, DIVER can enhance the recall of generated outputs by using PMI scores for re-ranking, which is suitable for text summarization. For example, DIVER$_R$ achieves improvements of 0.95 and 0.82 in average ROUGE scores on SAMSum with 7B and 13B models, respectively. On the other hand, dialogue-response and story-generation tasks emphasize precision and diversity in outputs. DIVER increases average BLEU and Distinct scores, demonstrating its superiority in balancing precision and diversity in LLM decoding.

**Code Generation** We employ Code-LLaMA-Instruct to evaluate the effectiveness of DIVER on code generation. As shown in Table 2 and Table 3, Pass@1 of DIVER outperforms existing methods, respectively surpassing greedy search by 2.07 and 1.14 scores on 7B and 13B models. The results demonstrate that using the test code cases (a part of inputs) for verification will boost the reliability of code generation, resulting in more cases being passed.

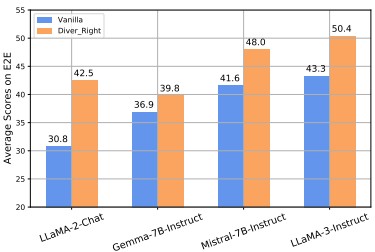

Figure 5: Performance improvements on E2E achieved by using DIVER$_R$ with various LLMs.

**Performance on other LLMs** We finally conducted experiments on various LLMs using the E2E dataset. As shown in Figure 5, DIVER obtains consistently enhanced performance with different LLMs. This demonstrates that DIVER is robust and effective across various LLMs.

## 5 ANALYSIS

### 5.1 DIVER IMPROVES FAITHFULNESS

DIVER is proposed to address the hallucination problem in LLMs, primarily focusing on enhancing the faithfulness of generated outputs. To accurately

assess the effectiveness of DIVER in this regard, we randomly selected 128 examples from the Flores Zh-En (Machine Translation) and E2E (Table-to-Text) test sets for human evaluation.

For Flores Zh-En, we ask annotators to choose the translation that is most faithful to the input from among the candidates produced by different decoding strategies, including greedy search, beam search (Freitag & Al-Onaizan, 2017), and DIVER. As shown in Figure 4 (a), DIVER provides the most faithful translations in 35.94% of the examples, outperforming both greedy search and beam search. For E2E, we instruct annotators to compare the outputs generated by DIVER with those produced by other decoding methods, judging which is more faithful. Figure 4 (b) indicates that DIVER achieves high win rates (48.44% $\sim$ 62.50%) in most cases.

## 5.2 NUMBER OF DIVERGENCE POINTS, SPAN LENGTH AND HYPER-PARAMETER $\gamma$

**Number of Divergence Points**  Figure 6 (a) illustrates the average number of divergence points per example across various tasks. We observe that tasks with deterministic outputs, like code generation (MBPP) and translation (Flores Ar-En), typically have fewer divergence points. In contrast, tasks with greater output variability, such as SAMSum and ROCStory, exhibit a higher number of divergence points.

**Span Length**  Figure 6 (b) illustrates the distribution of span lengths across various tasks. DIVER-RIGHT employs adaptive methods to derive dynamic spans, resulting in varied span lengths. For instance, in MBPP, span lengths exhibit a broader range from 0 to 60, with an average length of 14.9. Conversely, the span lengths in ROCStory and E2E are more tightly clustered between 0 and 20, with average lengths of approximately 4. This highlights DIVER's capability to provide spans of appropriate lengths for verification, consequently enhancing performance automatically. DIVER-LEFT generates shorter spans but maintains similar patterns across various tasks, just like DIVER-RIGHT. Thus, we do not elaborate further on DIVER-LEFT.

**Influence of $\gamma$**  Figure 6 (c) shows the impact of $\gamma$ on performance enhancements (subtracting the baseline performances) across various tasks on development sets. The most significant improvements are consistently observed when $\gamma \leq 0.3$ across all tasks. However, subtle variations exist among tasks. For Flores Ar-En and ROCStory, setting $\gamma = 0.1$ yields optimal results, whereas for E2E, MBPP and SAMSum, $\gamma = 0.3$ proves most effective. Nevertheless, all values of $\gamma$ lead to improvements. The analysis underscores the recommendation to opt for $\gamma \leq 0.3$ in practical deployment.

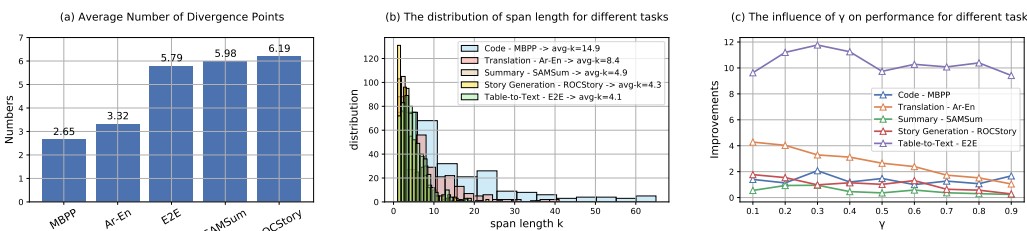

Figure 6: The analyses about the number of divergence points, length of dynamic spans, and the influence of $\gamma$ on development sets.

## 5.3 DECODING SPEED AND ACCELERATION

Decoding speed is the limitation of DIVER, which is hindered by the additional computation required for verification steps. Table 4 shows the performance and speed of various decoding methods. Compared to vanilla decoding methods such as greedy search or nucleus sampling, all recently proposed techniques demonstrate slower speeds. CAD necessitates double computation at each decoding step, making it the slowest among them. DIVER conducts verification at divergence points, maintaining a better speed than CAD but still lagging behind vanilla decoding. Conversely, CD utilizes a smaller model for contrastive decoding, resulting in faster speeds.

| Model | Decoding Method | E2E | Flores Ar-En | ROCStory | SAMSum | Speed (tokens/s) |
|-------|-----------------|-----|--------------|----------|--------|------------------|
| **7B** | Vanilla | 30.75 | 25.43 | 37.56 | 28.87 | 38.91 (1.00 ×) |
| | $\text{CD - CONTRAST}_{1.1B}$ | 30.29 | 25.33 | 37.78 | 28.32 | 33.08 (0.85 ×) |
| | CAD | 34.60 | 27.10 | 38.28 | 29.49 | 20.08 (0.51 ×) |
| | $\text{DIVER}_R \text{ - VERIFY}_{7B}$ | **42.52** | 28.15 | 38.54 | 29.82 | 24.49 (0.63 ×) |
| | $\text{DIVER}_R \text{ - VERIFY}_{1.1B}$ | 42.19 | **29.06** | **38.73** | **30.13** | 32.87 (0.84 ×) |
| **13B** | Vanilla | 34.57 | 30.27 | 37.51 | 30.05 | 27.36 (1.00 ×) |
| | $\text{CD - CONTRAST}_{1.1B}$ | 35.24 | 29.37 | 37.88 | 29.69 | 23.85 (0.87 ×) |
| | CAD | 39.08 | 32.68 | 38.24 | 30.78 | 15.13 (0.55 ×) |
| | $\text{DIVER}_R \text{ - VERIFY}_{13B}$ | **48.87** | **34.15** | 38.84 | 30.87 | 16.69 (0.61 ×) |
| | $\text{DIVER}_R \text{ - VERIFY}_{1.1B}$ | 48.22 | 32.53 | **38.90** | **31.19** | 22.98 (0.84 ×) |

Table 4: The comparison of performance and speed among different decoding methods with LLaMA-2-7B-Chat.

Drawing inspiration from this, we also utilize Tiny-LLaMA-1.1B-Chat as the verification model ($\text{DIVER}_R$ - $\text{VERIFY}_{1.1B}$). Compared to $\text{DIVER}_R$ using the same model for verification, $\text{DIVER}_R$ - $\text{VERIFY}_{1.1B}$ significantly boosts decoding speed. Interestingly, using small models for verification only marginally decreases performance, sometimes even yielding better improvements, making it conducive to practical deployment.

## 6 RELATED WORK

Recently, large language models (LLMs) have emerged as the predominant focus of research, primarily owing to their capacity to adeptly tackle a wide range of natural language processing tasks (Brown et al., 2020; Ouyang et al., 2022). Nonetheless, as LLMs are not tailored for specific downstream tasks, they often encounter challenges such as generating unfaithful outputs or factual inaccuracies, a phenomenon commonly referred to as hallucination problems (Rawte et al., 2023; Ji et al., 2023; Huang et al., 2023b).

Various decoding methods are proposed to mitigate this issue. To relieve the factual errors (Maynez et al., 2020; Huang et al., 2023a), Li et al. (2023) propose contrastive decoding, employing the difference between the distributions of LLMs and the corresponding weaker model for token selection. Chuang et al. (2024) calculate the token distribution contrasting the logits difference between the last layer and a premature layer. Xu et al. (2024) adopt multiple LLMs for reliable inference.

Recent studies have endeavored to address the challenge of inconsistency by ensuring contextual coherence during inference. van der Poel et al. (2022) and Shi et al. (2023) advocate adjusting the output distribution by reducing reliance on prior context knowledge. In previous studies on attribute-controlled text generation, Yang & Klein (2021) and Krause et al. (2021) employ Bayesian factorization, requiring each predicted token to accurately predict associated attributes. This methodology is further applied in LLM decoding, as demonstrated by (Tu et al., 2023).

Regrettably, the effectiveness of the aforementioned faithful decoding methods cannot be guaranteed for various tasks, particularly when the input $x$ is information-rich. As discussed in section A.2, the substantial variance in information content between $x$ and the individual token $y_i$ poses a challenge. DIVER tackles this issue by implementing adaptive token spans for PMI verification, thereby enhancing LLM decoding both in the performance and versatility across different tasks.

## 7 CONCLUSION AND FUTURE WORK

In this work, we propose DIVER to enhance the large language model decoding through span-level point-wise mutual information verification. Experimental results on various downstream tasks demonstrate the effectiveness of our method. Extensive analyses reveal the characteristics of DIVER, highlighting both its advantages and disadvantages, as well as the alleviation strategy. Future work will focus on combining DIVER with speculative decoding (Stern et al., 2018; Xia et al., 2023; Leviathan et al., 2023) to accelerate inference for LLMs.

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

## A  APPENDIX

### A.1  THE USE OF LARGE LANGUAGE MODELS

In this work, large language models are used to improve writing, mainly for text polishing and typo correction.

### A.2  WHY USE TOKEN SPANS FOR VERIFICATION

One of the primary innovations of this study lies in the utilization of token spans for PMI calculation. This section addresses the rationale behind our preference for spans over individual tokens in verification.

|                        | E2E   | Zh-En | MBPP  | ROCStory | SAMSum |
|------------------------|-------|-------|-------|----------|--------|
| Vanilla                | 30.75 | 39.14 | 46.60 | 37.56    | 28.87  |
| Beam Search            | 37.52 | 39.76 | **49.80** | 37.11 | 29.34  |
| BAYESIAN (Tu et al., 2023) | 39.95 | 39.33 | 46.20 | 38.16 | 28.73  |
| DIVER-TOKEN            | 41.25 | 39.96 | 47.33 | 38.16    | 29.34  |
| DIVER$_R$              | **42.52** | **40.77** | 48.67 | **38.54** | **29.82** |

Table 5: The comparison of methods that employ a single token or token spans to perform verification during decoding with LLaMA-2-7B-Chat.

| Input | name : The Punter \| Type : pub \| food : English \| price : high \| area : riverside \| family friendly : yes \| near : Raja Indian Cuisine |
|-------|-------|
| **Greedy** | The Punter is a riverside pub offering high-quality English food in a family-friendly atmosphere. |
| **CD** | The Punter is a riverside pub offering high-quality English food in a family-friendly atmosphere, but it does not cater to families. |
| **CAD** | The Punter is a riverside pub offering high-quality English food in a family-friendly atmosphere. |
| **BAYESIAN** | The Punter is a high-end English pub located on the riverside, offering a range of traditional dishes with a modern twist, and is family-friendly. |
| **DIVER** | The Punter is a riverside pub serving high-priced English food, with family-friendly atmosphere, located near Raja Indian Cuisine. |

Table 6: An example (E2E) that illustrates DIVER maintaining the integrity of semantics with span-level verification and thus avoiding the omission problem.

As illustrated in Table 5, the performance of DIVER$_R$, which employs span-level verification, consistently surpasses that of DIVER-TOKEN, which relies on single-token verification. This highlights the significance of sufficient information in ensuring accurate PMI calculation, thereby impacting the effectiveness of downstream tasks.

Furthermore, we conduct a comparative analysis between DIVER$_R$, beam search, and the BAYESIAN based decoding approach (Yang & Klein, 2021; Tu et al., 2023). Specifically, BAYESIAN is similar to DIVER-TOKEN, which also utilizes individual tokens for verification. The key differences are: DIVER-TOKEN uses the delta of input likelihood for verification when decoding $y_i$, while BAYESIAN directly predicts the input likelihood; (2) DIVER-TOKEN operates at divergence points, whereas BAYESIAN functions at each decoding step, similar to beam search. The results demonstrate that, compared to beam search and BAYESIAN, DIVER$_R$ exhibits superior versatility, yielding notable enhancements across multiple tasks.

Besides demonstrating superior performance, we use a specific example picked from E2E (table-to-text) to illustrate how DIVER addresses the omission problem and thereby improves faithfulness. As shown in Table 6, when given a sequence of table elements as the input, LLaMA-2-7B-Chat with existing decoding strategies generates sentences that consistently ignore *near: Raja Indian Cuisine*.

In contrast, DIVER, which employs token spans for verification, provides sufficient information for span selection and successfully generates a sentence that includes this important element. This underscores the importance of employing spans with adequate information for effective verification.

## A.3 SUPPLEMENTARY EXPERIMENTS

We also conduct experiments on the instruction following task with the AlpacaEval (Dubois et al., 2023) dataset. We measure the pairwise Win Rate against Text-Davinci-003 using GPT-4[7].

As shown in Table 7, we employ nuclear sampling as the baseline and compare its win rate to that of DIVER. The results demonstrate that DIVER is not only effective for traditional NLP tasks but also excels in instruction-following tasks (+7.45% for DIVER$_R$), which are crucial in the research of LLMs[8].

| Decoding | Sampling | DIVER$_L$ | DIVER$_R$ |
|---|---|---|---|
| **Win Rate** | 58.14% | 63.11% | **65.59**% |

Table 7: Win rate of LLaMA-2-7B-Chat generations using different decoding methods against Text-Davinci-003.

## A.4 INSTRUCTION TEMPLATE

The instruction templates for each dataset are listed in Table 8-16. In our method, DIVER employs the same LLMs for PMI calculation, which need examples with backward instructions. The backward examples are also included in the corresponding tables.

| **PROMPT FOR E2E** |
|---|
| Main Components: `[INPUT]` |
| Write a Sentence to describe the Main Components. Sentence: |
| **BACKWARD EXAMPLE FOR DIVER** |
| Sentence: `[INCOMPLETE_OUTPUT]` |
| Extract the Main Components from the Sentence. Main Components: `[INPUT]` |

Table 8: Instruction and backward example for E2E.

| **PROMPT FOR TRANSLATION (FLORES-200)** |
|---|
| `[SOURCE]`: `[INPUT]` |
| Translate the `[SOURCE]` sentence into `[TARGET]` sentence. `[TARGET]`: |
| **BACKWARD EXAMPLE FOR DIVER** |
| `[TARGET]`: `[INCOMPLETE_OUTPUT]` |
| Translate the `[TARGET]` sentence into `[SOURCE]` sentence. `[SOURCE]`: `[INPUT]` |

Table 9: Instruction and backward example for Flores-200. `[SOURCE]` and `[TARGET]` refer to languages.

---

[7] *gpt-4-0613* API is employed for the evaluation

[8] Honestly speaking, evaluating using GPT-4 is somewhat expensive for us. So, we only assessed the three experiments listed in Table 7.

| **PROMPT FOR CNN/DAILYMAIL** |
| --- |
| Article: [INPUT] |
| Summarize the Article in one Sentence. Sentence: |
| **BACKWARD EXAMPLE FOR DIVER** |
| Summary: [INCOMPLETE_OUTPUT] |
| Expand the Summary to an Article. Article: [INPUT] |

Table 10: Instruction and backward example for CNN/DailyMail.

| **PROMPT FOR ROCSTORY** |
| --- |
| Four-Sentence-Story: [INPUT] |
| Write a Ending Sentence according to the given Four-Sentence-Story. Ending Sentence: |
| **BACKWARD EXAMPLE FOR DIVER** |
| Ending Sentence: [INCOMPLETE_OUTPUT] |
| Write a Four-Sentence-Story according to the given Ending Sentence. Four-Sentence-Story: [INPUT] |

Table 11: Instruction and backward example for ROCStory.

| **PROMPT FOR MBPP** |
| --- |
| You are an expert Python programmer, and here is your task: [TASK_DESCRIPTION] |
| Your code should pass these tests: |
| [TEST_CASE_1] |
| [TEST_CASE_2] |
| [TEST_CASE_3] |
| Your code should start with a [PYTHON] tag and end with a [/PYTHON] tag. |
| [PYTHON] |
| **BACKWARD EXAMPLE FOR DIVER** |
| You are an expert that can understand Python programs. Give you codes that start with a [PYTHON] tag and end with a [/PYTHON] tag. |
| [PYTHON] |
| [INCOMPLETE_OUTPUT] |
| [/PYTHON] |
| The above code should pass these tests: |
| [TEST_CASE_1] |
| [TEST_CASE_2] |
| [TEST_CASE_3] |

Table 12: Instruction and backward example for MBPP.

| **PROMPT FOR COMMONGEN** |
| --- |
| Given several concepts (*i.e.*, nouns or verbs), write a short and simple sentence that contains \*all\* the required words. The sentence should describe a common scene in daily life, and the concepts should be used in a natural way. |
| Concepts: [INPUT] |
| Sentence: |
| **BACKWARD EXAMPLE FOR DIVER** |
| Given a short and simple sentence, extract several concepts (i.e., nouns or verbs) from the sentence. |
| Sentence: [INCOMPLETE_OUTPUT] |
| Concepts: [INPUT] |

Table 13: Instruction and backward example for CommonGen.

| PROMPT FOR ALPACAEVAL |
| --- |
| [INPUT] |
| **BACKWARD EXAMPLE FOR DIVER** |
| [INCOMPLETE_OUTPUT] |
| Based on the response, the instruction can be: [INPUT] |

Table 14: Instruction and backward example for AlpacaEval.

| PROMPT FOR SAMSUM |
| --- |
| Dialogue: [INPUT] |
| Summarize the Dialogue in one Sentence. Sentence: |
| **BACKWARD EXAMPLE FOR DIVER** |
| Summary: [INCOMPLETE_OUTPUT] |
| Expand the Summary to a Dialogue. Dialogue: [INPUT] |

Table 15: Instruction and backward example for SAMSum.

| PROMPT FOR NATURAL QUESTIONS & WEB QUESTIONS |
| --- |
| Question: [$Q_1$] Answer: [$A_1$] \| Question: [$Q_2$] Answer: [$A_2$] \| $\cdots$ \| Question: [$Q_k$] Answer: [$A_k$] \| Question: [INPUT] Answer: |
| **BACKWARD EXAMPLE FOR DIVER** |
| Answer: [$A_1$] Question: [$Q_1$] \| Answer: [$A_2$] Question: [$Q_2$] \| $\cdots$ \| Answer: [$A_k$] Question: [$Q_k$] \| Answer: [INCOMPLETE_OUTPUT] Question: [INPUT] |

Table 16: $k$-shot prompt and backward prompt for Natural Question and Web Questions. We recommend using in-context-learning for unaligned models.

