# OpenReview forum: "DIVER : Large Language Model Decoding with Span-Level Mutual Information Verification"
_ICLR.cc/2026/Conference — Submitted to ICLR 2026_

### Official Review · Reviewer_3ADH · 2025-11-01

**Soundness:** 3
**Presentation:** 3
**Contribution:** 3
**Rating:** 4
**Confidence:** 4

**Summary:**

This paper introduces DIVER, an inference-time method designed to mitigate model hallucination. By identifying divergence points during decoding and concurrently computing PMI scores for the next k dynamic spans, the method selects tokens that are more faithful to the input, achieving better results across multiple tasks.

**Strengths:**

1.	Writing is clear and the experiments are comprehensive.

2.	Method is novel and achieves better performance than other decoding-time approaches.

**Weaknesses:**

1.	Practicality concerns. DIVER decode several candidate spans in parallel and invoke the LLM for every PMI computation, incurring substantial latency. As Table 4 shows, speed drops to roughly 60 % of vanilla decoding, yet Table 3 reveals only marginal gains on many benchmarks, raising questions about the cost-effectiveness of the method in real-world deployments.

2.	The choice of span length requires further investigation. Table 3 shows that DIVER_R outperforms DIVER_L, indicating that richer information is more important than considering more divergence points. However, the authors did not justify the rationale for defining the Dynamic Span based on the first occurrence of a risk point (either left or right). The impact of SPAN length and the number of included divergence points on performance remains to be further explored.

**Questions:**

1.	Could you report performance under equal computation (FLOPs)? For example, compare the BoN from two vanilla random samples with DIVER.

2.	Could you demonstrate the effectiveness of your dynamic span method? For instance, plot how performance changes as the span length increases and as the number of skipped risk points grows.

---

### Official Review · Reviewer_1qYx · 2025-11-01

**Soundness:** 1
**Presentation:** 1
**Contribution:** 2
**Rating:** 2
**Confidence:** 3

**Summary:**

This paper proposes DIVER, a novel decoding method for large language models. During decoding, DIVER first identifies a divergence point and then selects a span that scores high in mutual information as well as the likelihood. The effectiveness of the proposed method is shown in diverse datasets.

**Strengths:**

- Strong empirical performance
- Slower inference speed can be mitigated by using a smaller verifier. This is a very interesting observation.
- Extensive experiments are provided.

**Weaknesses:**

The biggest limitation of this manuscript is its writing and lack of clarity. I believe the manuscript requires significant rewriting and may need another round of peer review.

- The use of Point-wise Mutual Information (PMI) is poorly motivated. I am not convinced why or how Equation 2, which adds the PMI score to the logits, would improve the decoding process.
- PMI is not properly defined. It is currently defined implicitly in Equation 6. However, since PMI plays a central role in this paper, it deserves paragraphs dedicated to its definition and discussion of its characteristics.
- In computing PMI (Equations 6 and 7), the probability p(x|y) needs to be obtained, but it is unclear how this quantity is computed. Since an LLM only models p(y|x), this probability is difficult to compute. Although footnote 2 comments on this issue, it does not clarify how the probability is calculated. Additionally, the term “backward teacher-forcing decoding” is undefined.
- The symbol “~” is used in an unusual way (in Equations 1 to 6). Typically, “A ~ B” denotes that the random variable A is sampled from a distribution B. However, in Equations 1–6, the right-hand side is not a proper distribution.
- The description of DIVER in Section 3.2 is convoluted, and I do not think a practitioner could reproduce the method by reading this section. Providing an explicit algorithm would help, for example.

**Questions:**

See weaknesses

---

### Official Review · Reviewer_rZ8W · 2025-11-04

**Soundness:** 2
**Presentation:** 2
**Contribution:** 2
**Rating:** 4
**Confidence:** 4

**Summary:**

The paper proposes DIVER, which is a decoding method utilizing span-level pointwise mutual information (PMI) verification. DIVER uses token probability information to identify divergence steps, generates candidate steps and computes the PMI scores by assessing the log-likelihood gains of the input if the candidate spans are generated. The optimal span is selected based on the PMI re-ranked output distributions. DIVER also uses an adaptive method for obtaining token spans with dynamic lengths along the generation. The paper presents experiment results across several task domains to demonstrate the performance improvement of DIVER.

**Strengths:**

1. The proposed algorithm in the paper is well explained, effectively using visual examples.
2. The experiment was conducted across several task domains to present DIVER's effectiveness.
3. The paper also includes analyses regarding the potential limitation of the proposed algorithm.

**Weaknesses:**

1. Some crucial details of the algorithm seem to be missing, such as how many candidate spans were generated when the algorithm encounters a divergence point.
2. If DIVER is generating several candidate spans per divergence point, the experiment results could have been more convincing if the comparison was also done against baseline algorithms that also generate several candidate responses or partial sequences.
3. The experiment results do not include information such as standard deviation, which is crucial for the credibility of the results.

**Questions:**

## Questions
1. Could you let us know how many example spans were generated during the experiments?

## Suggestions
1. I think the caption of Figure 1 should be polished.
2. Figure 3 caption: I think `Bleu` should be `Blue`.
3. I think images in Figure 4, 5, 6 should be bigger. The paper might be able to save a few lines by polishing the main text.
4. Table 12: `Give you codes that start with a ...` does not seem to be a correct sentence.
5. Table 12: Could you have a look if the given prompt was copied correctly into the appendix?

---

### Meta-Review · Area_Chair_7Udb · 2026-01-06

**Summary:**

The paper proposes DIVER, a decoding method that detects divergence points, enumerates candidate spans, and re-ranks spans using span-level PMI-style verification (via input log-likelihood gains). The reviews are mixed (4/2/4). The main concerns driving the decision are (i) unclear and potentially flawed probabilistic formulation of PMI, (ii) insufficient algorithmic detail for reproducibility (e.g., number of candidate spans, exact procedure), and (iii) practicality: notable latency increase for often modest gains, plus missing statistical reporting.

**Reviewer Concerns:**

No author rebuttal/discussion content is present in the provided forum snapshot, so I cannot verify that any concerns were resolved.

**Reviewer Scores:**

Given the (apparent) absence of rebuttal/discussion and the strength of the core concerns, I expect little review score changes.

---

### Decision · Program_Chairs · 2026-01-26

Reject